# Effectiveness of Different Types of Core Decompression in Early-Stage Osteonecrosis of the Femoral Head: A Systematic Review and Meta-Analysis

**DOI:** 10.3390/medsci13040258

**Published:** 2025-11-03

**Authors:** Wojciech Konarski

**Affiliations:** Medical Rehabilitation Center, Sobieskiego 47D, 05-120 Legionowo, Poland; wkonarski@poczta.onet.pl; Tel.: +48-502110863

**Keywords:** osteonecrosis, decompression, bone graft, mesenchymal stem cells, platelet-rich plasma

## Abstract

Background: Osteonecrosis of the femoral head is a progressive disorder leading to femoral head collapse and early disability, often affecting young adults. Core decompression (CD) is the most established hip-preserving treatment for early-stage disease, yet the comparative benefits of biological and structural augmentation remain uncertain. Methods: This systematic review and meta-analysis, registered in PROSPERO (CRD420251108396), evaluated 14 studies encompassing 1210 patients treated with CD alone or CD combined with biological (e.g., platelet-rich plasma, mesenchymal stem cells, bone marrow aspirate concentrate) or structural (e.g., bone grafting, fibular support) augmentation. Results: Pooled random-effects models demonstrated that biological augmentation yielded significant improvements in Harris Hip Score and pain reduction (VAS) up to 24 months, with early peaks and subsequent stabilization, whereas structural augmentation showed no functional advantage at any time point. Radiological progression and conversion to total hip arthroplasty were not significantly reduced, though long-term trends favored biologically augmented CD. Conclusions: Overall, biological augmentation provides durable functional and symptomatic benefits in early-stage osteonecrosis, supporting its use as a first-line adjunct to CD, while structural augmentation appears less consistent and warrants further evaluation through large, standardized trials.

## 1. Introduction

Osteonecrosis of the femoral head—also known as avascular necrosis (AVN)—is a pathological condition in which osteocytes die due to loss of blood supply. This process can occur after direct trauma (such as a femoral neck fracture or hip dislocation that disrupts the arterial circulation) or from non-traumatic causes. Cumulative corticosteroid dose and chronic alcohol abuse are well-established risk factors for atraumatic osteonecrosis of the hip [1,2]. Regardless of etiology, the interruption of perfusion to the femoral head leads to bone marrow ischemia and osteocyte death. The necrotic bone gradually loses structural integrity; microfractures accumulate and, under the stress of weight-bearing, the subchondral bone eventually collapses, causing flattening of the femoral head and arthritic degeneration of the hip joint [1,3,4].

Femoral head osteonecrosis has a high impact on younger adults. The mean age at presentation is around the late 30s, and many patients are in their third to fifth decade of life [2]. Each year in the United States, an estimated 20,000–30,000 individuals are newly diagnosed with osteonecrosis of the femoral head [5]. In the United Kingdom, the condition represents the third leading indication for total hip arthroplasty (THA) among patients younger than 50 years [6]. These observations underscore the importance of early diagnosis and intervention before structural collapse occurs.

Early-stage (pre-collapse) osteonecrosis of the femoral head can be treated by various joint-preserving treatment strategies which aim to prevent disease progression and maintain the native hip. These range from conservative measures to operative interventions. Surgical interventions have a more established role in pre-collapse osteonecrosis, and they are routinely considered when lesions are detected at an early stage. A variety of hip-preserving surgical techniques have been employed, including core decompression (CD), bone grafting procedures (non-vascularized or vascularized grafts), and proximal femoral osteotomies. Among these options, CD is one of the most performed first-line procedures for early-stage osteonecrosis [1,7].

CD involves drilling one or more channels into the necrotic area of the femoral head to relieve the elevated intramedullary pressure and to create a conduit for new blood vessel in-growth. By decompressing the bone and marrow, this procedure is thought to reduce venous stasis and bone edema while promoting revascularization and osteogenesis in the necrotic zone. Clinically, CD can result in significant pain relief and functional improvement, particularly in small or medium-sized lesions before subchondral collapse has occurred. It is a relatively low-morbidity, cost-effective intervention and has therefore become a popular choice as an initial surgical treatment for early osteonecrosis [1,7].

To enhance the results of CD, numerous adjunctive techniques have been introduced in recent years. The rationale behind augmentation is either to amplify the biological healing potential of the decompression or to provide structural support to the weakened femoral head. Some investigators have used biologic agents and cell therapy in conjunction with CD. For example, autologous bone marrow aspirate concentrate (BMAC), which is rich in mesenchymal stem cells (MSCs) and growth factors or platelet-rich plasma (PRP), can be injected into the decompression track to promote new bone formation and angiogenesis in the necrotic area [7,8,9,10,11,12,13].

Overall, the optimal management strategy for early-stage osteonecrosis of the femoral head remains unclear. There is a broad spectrum of available interventions—from simple CD alone to a variety of augmentation techniques involving cells, biologics, grafts, or implants [2]. High-level comparative evidence is limited, and interpreting the existing literature is challenging due to heterogeneity in study designs, patient populations, and endpoints Considering this uncertainty, we undertook a systematic review and meta-analysis to compare the effectiveness of different types of core decompression in early-stage osteonecrosis of the femoral head. The aim of this study was to critically evaluate whether modern augmented CD techniques offer superior clinical outcomes and femoral head preservation compared to the traditional CD alone in patients with precollapse osteonecrosis of the femoral head.

## 2. Materials and Methods

### 2.1. Study Design

This systematic review and meta-analysis were conducted in accordance with the Preferred Reporting Items for Systematic Reviews and Meta-Analyses (PRISMA) guidelines [14]. The study protocol was prospectively registered in the PROSPERO international database (ID: CRD420251108396), ensuring methodological transparency and adherence to predefined objectives and procedures.

A structured approach was employed to evaluate treatment effects while accounting for clinical heterogeneity and temporal variations. Analyses were conducted by predefined groups (biological augmentation and structural augmentation) to provide an overview of overall efficacy and indirect comparisons between main groups to assess relative superiority) [15]. Results were stratified by follow-up time points (e.g., 3, 6, 12, 24, 36, 48, and 60 months) where data permitted, enabling examination of effect durability over time.

The current structure aligned with established meta-analytic principles, facilitating logical progression from broad to granular insights while minimizing redundancy [16]. Meta-analyses were conducted only for time points and groupings with at least two studies, as fewer pooling lacks statistical robustness and risks overinterpretation [17]. Single-study data were excluded from pooling to ensure replication and reduce bias [14]. Sensitivity analyses assessed impacts of study quality or outliers. All procedures used a significance level of α = 0.05.

### 2.2. Search Strategy

A comprehensive literature search was conducted by two independent reviewers in PubMed, Embase, Web of Science, the Cochrane Library, and ClinicalTrials.gov for studies published between 1 January 2000 and 31 July 2025. The search strategy combined controlled vocabulary (e.g., MeSH and Emtree terms) and free-text keywords related to osteonecrosis of the femoral head and core decompression. Detailed search strings adapted to the syntax of each database are provided in the Appendix A.

### 2.3. Eligibility Criteria

Studies were eligible if they included patients with osteonecrosis of the femoral head in ARCO stage I–II or equivalent classifications. Eligible interventions comprised core decompression performed either alone or in combination with bone grafting, stem cell augmentation, or other adjunctive techniques, with comparators restricted to core decompression alone. The outcomes of interest included changes in the Harris Hip Score (HHS), Visual Analog Scale (VAS), radiological progression and conversion to THA. We considered randomized controlled trials as well as prospective and retrospective cohort studies. Only articles published in English or Polish were included.

### 2.4. Study Selection

All records retrieved from the database search were imported into a reference management software (EndNote 20), and duplicates were removed. Titles and abstracts were screened to exclude studies that clearly did not meet the eligibility criteria. The full texts of potentially relevant articles were then assessed in detail. The process of study selection is illustrated in the PRISMA flow diagram.

### 2.5. Data Extraction

Data were extracted by two researchers using a standardized extraction form using Microsoft Excel. The following information was collected from each eligible study: first author, year of publication, country, study design, sample size, patient characteristics including age, and disease stage, details of the intervention and comparator, duration of follow-up, and reported outcomes (radiological progression, conversion to THA, HHS, VAS). Where studies reported multiple intervention arms, relevant data were extracted for each comparison. Any discrepancies in data extraction were discussed and resolved by consensus.

### 2.6. Risk of Bias and Certainty Assessment

For randomized controlled trials, we applied the Cochrane Risk of Bias 2.0 tool (RoB 2), evaluating domains such as the randomization process, deviations from intended interventions, missing outcome data, outcome measurement, and selection of the reported results. For non-randomized cohort studies, methodological quality was appraised using the Newcastle–Ottawa Scale (NOS), which addresses selection of study groups, comparability of cohorts, and outcome assessment. Discrepancies between reviewers were resolved through discussion and consensus.

The certainty of evidence for each outcome was evaluated according to the GRADE (Grading of Recommendations Assessment, Development and Evaluation) approach [18]. Evidence from randomized controlled trials was initially rated as high, and evidence from observational studies as low. The certainty was then downgraded based on five domains: risk of bias, inconsistency, indirectness, imprecision, and publication bias. The overall certainty of evidence was summarized as high, moderate, low, or very low. The assessment was performed qualitatively, following the GRADE Working Group guidelines, and is presented in the Summary of Findings table.

### 2.7. Statistical Analysis

Continuous outcomes, including the Harris Hip Score (HHS) for functional assessment and the Visual Analog Scale (VAS) for pain intensity, were treated as numeric variables. VAS scores were consistently reported on a 0–10 scale across all included studies, permitting direct pooling via mean differences (MDs). Pooled MDs with 95% confidence intervals (CIs) were estimated using post-treatment values—owing to comparable baseline characteristics across groups and the absence of change-from-baseline or adjusted data—through random-effects models, selected to account for expected heterogeneity in patient profiles and interventions [19]. These models incorporated inverse-variance weighting, with between-study variance (τ^2^) estimated via restricted maximum likelihood (REML) to enhance accuracy in small samples [16]. Results encompassed the Z-test statistic and *p*-value for overall effect significance, supplemented by prediction intervals in cases of substantial heterogeneity.

Binary outcomes encompassed radiological progression (worsening on imaging) and conversion to THA. Pooled risk ratios (RRs) with 95% CIs were estimated using random-effects models, selected for interpretability in risk contexts and to accommodate heterogeneity [20]. Inverse-variance weighting was applied, with τ^2^ estimated via REML; continuity corrections (adding 0.5 to zero-event cells) mitigated bias in sparse data [21]. Results included the Z-test statistic and *p*-value for overall effect, plus the number needed to treat (NNT) for clinical interpretation where applicable. As a sensitivity analysis, we employed a generalized linear mixed model (GLMM) with a logit link to estimate pooled odds ratios (ORs). The significance of heterogeneity was assessed using both the Wald test and likelihood ratio test (LRT), providing complementary evaluations of between-study variance. For multi-arm studies with a shared control, the control group events and sample size were split proportionally among intervention arms to avoid double-counting and unit-of-analysis errors.

Heterogeneity among studies was evaluated using the Cochrane Q-test, which provides a Chi^2^ statistic and associated *p*-value. The I^2^ statistic was employed to quantify the proportion of total variability attributable to between-study differences, with thresholds interpreted as low (50%). Additionally, tau-squared (τ^2^) was calculated to estimate between-study variance, accompanied by 95% confidence intervals derived via the Q-profile method [22]. In instances of substantial heterogeneity, potential sources were investigated through subgroup analyses, although meta-regression was precluded due to the limited number of studies per analysis [23].

No formal adjustment for multiplicity was applied, given the exploratory nature and interdependence of outcomes; results were interpreted holistically, focusing on clinical relevance rather than *p*-value thresholds.

Differences between subgroups were assessed using the Chi^2^ interaction test within random-effects models, which partitions heterogeneity into within- and between-subgroup components to evaluate if pooled effect estimates vary significantly across categories [19] with *p* < 0.05 indicating evidence of differential effects [20].

Analyses were conducted using the R Statistical language with dedicated packages for meta-analyses (version 4.3.3; R Core Team, 2024).

## 3. Results

The meta-analysis incorporated data from 14 unique studies, and the detailed literature inclusion and exclusion process is shown in the flow chart (Figure 1). Included studies primarily originated from China (*n* = 6), with additional contributions from Europe (Belgium, France, Germany, the Netherlands, the United Kingdom; *n* = 4), the Middle East (Egypt, Iraq, Turkey; *n* = 3), and India (*n* = 1). Study designs included randomized controlled trials (RCTs; *n* = 4), retrospective analyses (*n* = 6), prospective comparative or cohort studies (*n* = 3), and one randomized prospective study. Interventions predominantly involved biological augmentation (e.g., platelet-rich plasma, mesenchymal stem cells, bone marrow concentrate; *n* = 10 studies) or structural augmentation (e.g., bone grafting techniques; *n* = 4 studies), compared against core decompression alone. Participant numbers ranged from 18 to 263 per study (total N = 1210 across treatment and control arms), with a mean age of 39.1 years (range: 32.7 to 48.4 years), male predominance (mean = 74.6%, range: 51.5% to 100%,), and follow-up durations varying from 11.3 to 64.3 months. Table 1 presents detailed characteristics of included studies.

### 3.1. Effects of Biological Augmentation on HHS Scores

The present meta-analysis examined the efficacy of biological augmentation strategies compared to control interventions in improving HHS. Analyses were conducted at discrete follow-up intervals of 6, 12, and 24 months post-intervention, drawing from 3 to 4 studies per time point (total participants: 112 at 6 months, 242 at 12 months, 407 at 24 months).

At the 6-month follow-up, biological augmentation yielded a statistically significant and clinically substantial improvement in HHS relative to control, with a pooled MD of 8.39 (95% CI: 5.63 to 11.15; Z = 5.97; *p* < 0.001; Figure 2). Heterogeneity was absent (I^2^ = 0.0%; τ^2^ = 0; Q = 0.59, df = 2, *p* = 0.744), indicating consistent effects across the three included studies [27,32,36]. This early benefit indicates rapid functional gains. By the 12-month follow-up, the advantage persisted but attenuated, reflected in a pooled MD of 3.65 (95% CI: 0.22 to 7.08; Z = 2.08; *p* = 0.0371; Figure 3). While the MD of 3.65 at 12 months achieved statistical significance, it did not exceed the MCID threshold of 8–10 points, indicating limited clinical relevance at this interval despite persistence of effect. Substantial heterogeneity was observed (I^2^ = 70.4%; τ^2^ = 7.86; Q = 10.15, df = 3, *p* = 0.017), which may stem from variations in study methodologies, patient demographics, or specific augmentation subtypes across the four studies [27,31,32,37]. This moderation in effect size implies a potential plateau or waning of initial therapeutic impacts. At 24 months, the treatment effect stabilized at a clinically meaningful level, with a pooled MD of 5.04 (95% CI: 2.26 to 7.81; Z = 3.56; *p* < 0.001; Figure 4). Moderate heterogeneity was present (I^2^ = 41.8%; τ^2^ = 3.52; Q = 5.16, df = 3, *p* = 0.161), revealing reasonable consistency among the four studies [28,30,33,37].

### 3.2. Effects of Structural Augmentation on HHS Scores

The meta-analysis assessed the impact of structural augmentation strategies versus control on HHS. Evaluations were performed at 12, 24, and 48 months post-intervention, based on 2 to 3 studies per time point (total participants: 597 at 12 months, 255 at 24 and 48 months).

At the 12-month follow-up, structural augmentation showed no significant difference in HHS compared to control, with a pooled MD of −2.53 (95% CI: −7.67 to 2.62; Z = −0.96; *p* = 0.336; Figure 5). Extreme heterogeneity was evident (I^2^ = 98.4%; τ^2^ = 20.29; Q = 125.74, df = 2, *p* < 0.001), likely influenced by variations in grafting techniques or patient cohorts across the three studies [24,25,31]. The negative MD indicates a potential trend toward inferior outcomes with augmentation, though the wide CI precludes definitive conclusions.

By the 24-month follow-up, the lack of benefit persisted, with a pooled MD of −2.29 (95% CI: −13.34 to 8.75; Z = −0.41; *p* = 0.684; Figure 6). Pronounced heterogeneity was observed (I^2^ = 98.6%; τ^2^ = 62.62; Q = 71.38, df = 1, *p* < 0.001), reflecting methodological disparities in the two studies [24,25]. This non-significant result, coupled with the overlapping CI, indicates no discernible long-term functional advantage.

At 48 months, structural augmentation again demonstrated no significant effect on HHS, yielding a pooled MD of −2.73 (95% CI: −13.67 to 8.22; Z = −0.49; *p* = 0.625; Figure 7). Heterogeneity remained substantial (I^2^ = 98.6%; τ^2^ = 61.50; Q = 70.54, df = 1, *p* < 0.001), underscoring persistent variability in the same two studies [24,25].

**Figure 5 medsci-13-00258-f005:**
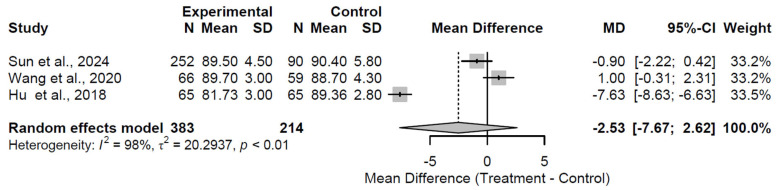
Forest plot of pooled mean differences in Harris Hip Score for structural augmentation versus control at 12 months follow-up [24,25,31].

**Figure 6 medsci-13-00258-f006:**
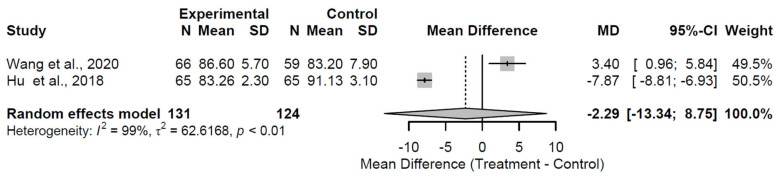
Forest plot of pooled mean differences in Harris Hip Score for structural augmentation versus control at 24 months follow-up [24,25].

**Figure 7 medsci-13-00258-f007:**
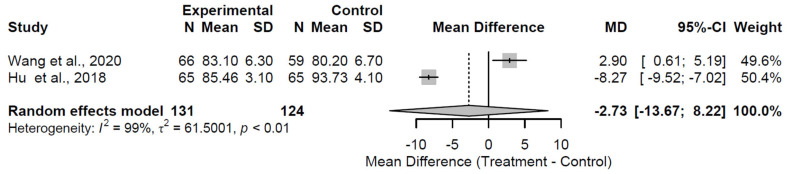
Forest plot of pooled mean differences in Harris Hip Score for structural augmentation versus control at 48 months follow-up [24,25].

### 3.3. Effects of Structural Augmentation on HHS Scores (Biological Augmentation vs. Structural Augmentation)

This analysis conducted indirect comparisons between biological augmentation and structural augmentation versus control on HHS. Comparisons utilized combined random-effects models treating main groups as subgroups, with interaction tests for differential effects, at shared follow-up points of 12 and 24 months. At 12 months, 7 studies were included (4 biological, 3 structural; total participants: 839); at 24 months, 6 studies (4 biological, 2 structural; total participants: 662).

At the 12-month follow-up, the overall pooled effect across groups was non-significant (MD = 0.92, 95% CI: −2.80 to 4.64; Z = 0.48; *p* = 0.628; Figure 8), with extreme heterogeneity (I^2^ = 96.6%; τ^2^ = 22.37; Q = 176.82, df = 6, *p* < 0.001), likely driven by diverse augmentation mechanisms and study designs. Subgroup-specific estimates revealed a modest benefit for biological augmentation (MD = 3.65) versus a non-significant deficit for structural (MD = −2.53), with the interaction test approaching significance (Q = 3.83, df = 1, *p* = 0.050), indicating potential superiority of biological approaches at this interval, though tempered by high within-group variability.

By the 24-month follow-up, the combined effect remained non-significant (MD = 2.67, 95% CI: −2.13 to 7.46; Z = 1.09; *p* = 0.275; Figure 9), accompanied by pronounced heterogeneity (I^2^ = 97.3%; τ^2^ = 32.13; Q = 184.77, df = 5, *p* < 0.001). Biological augmentation maintained a positive trend (MD = 5.04), while structural augmentation showed no advantage (MD = −2.29); however, the interaction test was non-significant (Q = 1.59, df = 1, *p* = 0.207), indicating no robust evidence of differential efficacy between groups at this extended time point.

**Figure 8 medsci-13-00258-f008:**
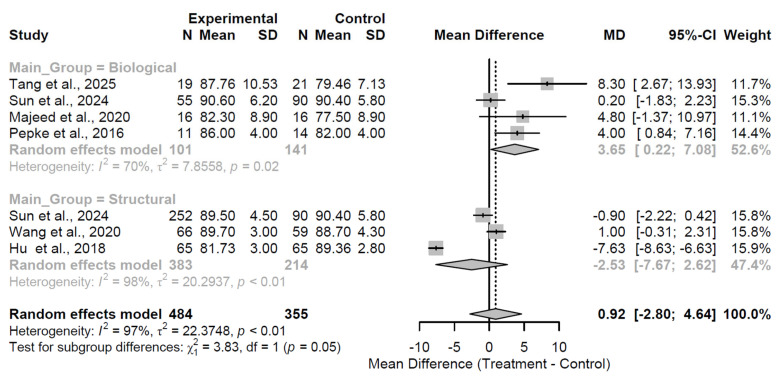
Forest plot of pooled mean differences in Harris Hip Score for biological versus structural augmentation at 12 months follow-up [24,25,27,31,32,37].

**Figure 9 medsci-13-00258-f009:**
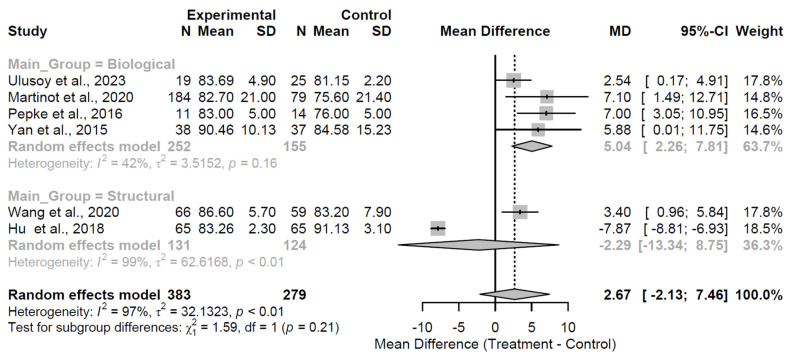
Forest plot of pooled mean differences in Harris Hip Score for biological versus structural augmentation at 24 months follow-up [24,25,28,30,33,37].

### 3.4. Effects of Biological Augmentation on VAS Scores

The meta-analysis investigated the efficacy of biological augmentation strategies versus control on VAS scores. Assessments were performed at follow-up durations of 3, 6, 12, and 24 months, incorporating 2 to 4 studies per time point (total participants: 84 at 3 months, 116 at 6 months, 141 at 12 months, 69 at 24 months) with negative MDs indicating reduced pain favoring augmentation.

At the 3-month follow-up, biological augmentation did not significantly alleviate pain compared to control, with a pooled MD of −0.23 (95% CI: −0.58 to 0.12; Z = −1.29; *p* = 0.198; Figure 10). Heterogeneity was absent (I^2^ = 0.0%; τ^2^ = 0; Q = 0.86, df = 1, *p* = 0.354), reflecting consistency in the two studies [27,28]. The non-significant effect infers minimal early analgesic impact.

By the 6-month follow-up, a significant pain reduction emerged, evidenced by a pooled MD of −0.86 (95% CI: −1.39 to −0.33; Z = −3.18; *p* = 0.002; Figure 11). Substantial heterogeneity was present (I^2^ = 72.3%; τ^2^ = 0.17; Q = 7.21, df = 2, *p* = 0.027), possibly attributable to variations in augmentation subtypes or patient baselines across the three studies [27,28,32]. This marked improvement indicates accelerating therapeutic benefits, aligning with enhanced tissue repair and anti-inflammatory effects.

At 12 months, the benefit persisted significantly, though attenuated, with a pooled MD of −0.66 (95% CI: −0.89 to −0.43; Z = −5.57; *p* < 0.001; Figure 12). No heterogeneity was detected (I^2^ = 0.0%; τ^2^ = 0; Q = 0.89, df = 3, *p* = 0.8275), denoting uniform outcomes in the four studies [27,28,32,37]. The sustained yet reduced effect implies a stabilization of pain relief.

At the 24-month follow-up, biological augmentation continued to demonstrate significant pain mitigation, with a pooled MD of −0.50 (95% CI: −0.75 to −0.25; Z = −3.88; *p* < 0.001; Figure 13). Heterogeneity was negligible (I^2^ = 0.0%; τ^2^ = 0; Q = 0.00, df = 1, *p* = 1.000), supporting consistent findings in the two studies [28,37]. This enduring reduction affirms long-term analgesic efficacy, though the modest magnitude reveals diminishing incremental gains over extended periods.

**Figure 10 medsci-13-00258-f010:**
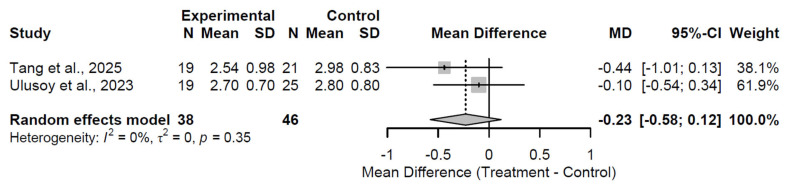
Forest Plot of pooled mean differences in Visual Analog Scale score for biological augmentation versus control at 3 months follow-up [27,28].

**Figure 11 medsci-13-00258-f011:**
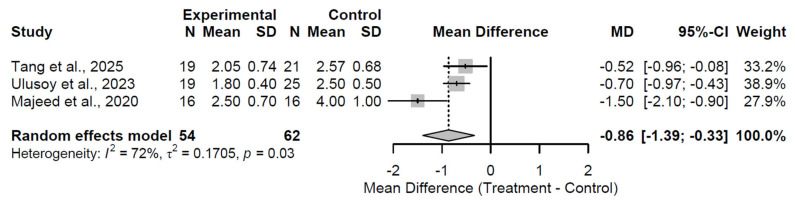
Forest Plot of pooled mean differences in Visual Analog Scale score for biological augmentation versus control at 6-month follow-up [27,28,32].

**Figure 12 medsci-13-00258-f012:**
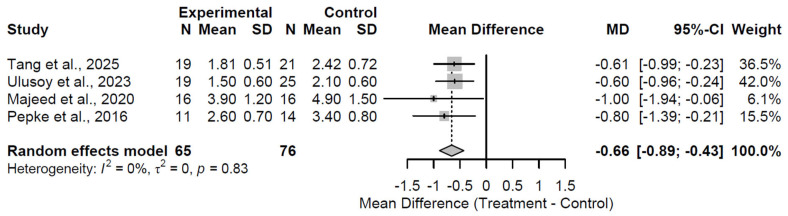
Forest Plot of pooled mean differences in Visual Analog Scale score for biological augmentation versus control at 12-month follow-up [27,28,32,37].

**Figure 13 medsci-13-00258-f013:**
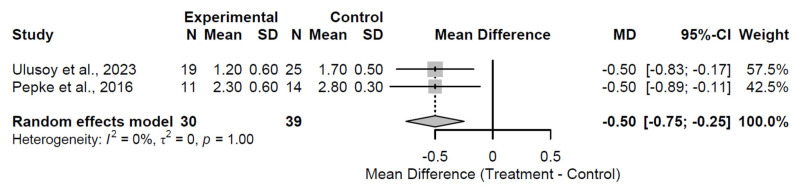
Forest Plot of pooled mean differences in Visual Analog Scale score for biological augmentation versus control at 24-month follow-up [28,37].

### 3.5. Effects of Biological Augmentation on Radiological Progression

The meta-analysis evaluated the efficacy of biological augmentation strategies versus control in mitigating radiological progression, defined as worsening on imaging scales (e.g., ARCO or Ficat staging) in patients with early-stage osteonecrosis of the femoral head. Analyses focused on time points with sufficient data (≥2 studies), specifically 12 months (3 studies; total hips: 127) and 60 months (2 studies; total hips: 494). Pooled RRs with 95% CIs were estimated, where RR < 1 indicates reduced progression risk favoring augmentation.

At the 12-month follow-up, biological augmentation did not significantly lower the risk of radiological progression compared to control, with a pooled RR of 0.58 (95% CI: 0.28 to 1.19; Z = −1.49; *p* = 0.137; Figure 14). Low heterogeneity was observed (I^2^ = 11.8%; τ^2^ = 0.06; Q = 2.27, df = 2, *p* = 0.322), inferring reasonable consistency across the three studies [27,29,32]. The non-significant effect, despite a trend toward risk reduction, implies limited early protective impact on disease advancement.

By the 60-month follow-up, a borderline significant reduction in progression risk emerged, with a pooled RR of 0.37 (95% CI: 0.14 to 1.02; Z = −1.93; *p* = 0.054; Figure 15). Moderate heterogeneity was present (I^2^ = 55.2%; τ^2^ = 0.36; Q = 2.23, df = 1, *p* = 0.135), possibly attributable to differences in patient cohorts or intervention specifics in the two studies [31,34]. This near-significant finding indicates emerging long-term benefits, with the lower RR indicating a potential halving of progression risk, though the upper CI approaching unity cautions against overinterpretation.

**Figure 14 medsci-13-00258-f014:**
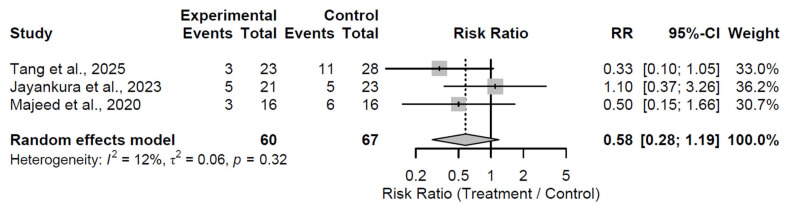
Forest plot of pooled risk ratios for radiological progression in biological augmentation versus control at 12-month follow-up [27,29,32].

**Figure 15 medsci-13-00258-f015:**
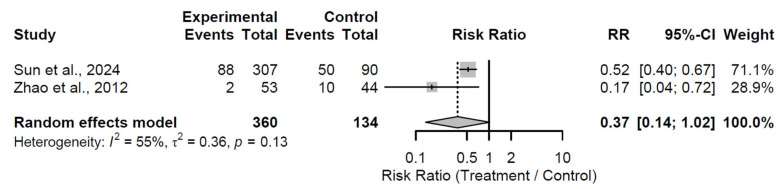
Forest plot of pooled risk ratios for radiological progression in biological augmentation versus control at 60-month follow-up [31,34].

As a sensitivity analysis, At the 12-month follow-up, biological augmentation demonstrated a non-significant trend toward reduced odds of radiological progression compared to control, with a pooled OR of 0.46 (95% CI: 0.20 to 1.05; Z = −1.84; *p* = 0.066). Heterogeneity was low (τ^2^ = 0; I^2^ = 18.9% [0.0% to 91.6%]; H = 1.11 [1.00 to 3.44]), with non-significant tests for heterogeneity (Wald-type Q = 2.47, df = 2, *p* = 0.291; likelihood ratio test Q = 2.52, df = 2, *p* = 0.283), indicating consistent effects across the included studies. By the 60-month follow-up, biological augmentation significantly lowered the odds of radiological progression relative to control, yielding a pooled OR of 0.29 (95% CI: 0.19 to 0.46; Z = −5.27; *p* < 0.001). Heterogeneity remained minimal (τ^2^ = 0; I^2^ = 8.3%; H = 1.04), confirmed by non-significant heterogeneity tests (Wald-type Q = 1.09, df = 1, *p* = 0.296; likelihood ratio test Q = 1.24, df = 1, *p* = 0.267), indicating high concordance between the two studies.

This GLMM-based sensitivity analysis aligns closely with the primary results, reinforcing a non-significant early trend and a stronger long-term protective effect against radiological progression, with the 60-month finding achieving statistical significance in the OR model (*p* < 0.001) compared to borderline significance in the RR model (*p* = 0.054). The consistent low heterogeneity across both analyses, along with similar directional effects despite the shift from RR to OR measures, underscores the robustness of the findings and enhances confidence in the emerging durability of biological augmentation benefits.

### 3.6. Effects of Structural Augmentation on Radiological Progression

The meta-analysis examined the efficacy of structural augmentation strategies versus control in preventing radiological progression, defined as deterioration on imaging classifications in patients with early-stage osteonecrosis of the femoral head. Analysis was restricted to the 36-month follow-up, as the sole time point with adequate data (≥2 studies), encompassing 2 studies and 177 hips (90 in treatment arms, 87 in control arms).

At the 36-month follow-up, structural augmentation did not significantly mitigate radiological progression compared to control, with a pooled RR of 0.95 (95% CI: 0.22 to 4.10; Z = −0.06; *p* = 0.949; Figure 16). Substantial heterogeneity was evident (I^2^ = 91.0%; τ^2^ = 1.01; Q = 11.14, df = 1, *p* = 0.001), potentially arising from differences in grafting procedures or patient selection criteria between the two studies [25,26].

The GLMM-based sensitivity analysis aligns with the primary results, demonstrating no protective effect of structural augmentation against radiological progression with a pooled OR of 1.17 (95% CI: 0.20 to 6.83; Z = 0.17; *p* = 0.864). The high heterogeneity (τ^2^ = 1.3531; τ = 1.1632; I^2^ = 91.4% [69.7% to 97.6%]; H = 3.41 [1.82 to 6.40]), supported by significant tests for heterogeneity as assessed by the Wald test (Q = 11.62, df = 1, *p* = 0.001) and likelihood ratio test (LRT; Q = 12.41, df = 1, *p* < 0.001), underscores potential differences in study characteristics or patient populations, limiting interpretability and emphasizing the need for additional research to clarify long-term outcomes.

### 3.7. Effects of Biological Augmentation on Conversion to THA

The meta-analysis appraised the efficacy of biological augmentation strategies versus control in preventing conversion to THA, a binary endpoint signifying treatment failure and surgical escalation in patients with early-stage osteonecrosis of the femoral head. Analysis was limited to the 24-month follow-up, the sole time point with sufficient data (≥2 studies), comprising 2 studies and 281 hips (194 in treatment arms, 87 in control arms).

At the 24-month follow-up, biological augmentation did not significantly decrease the risk of conversion to THA compared to control, with a pooled RR of 0.70 (95% CI: 0.44 to 1.10; Z = −1.54; *p* = 0.124; Figure 17). Minimal heterogeneity was observed (I^2^ = 1.1%; τ^2^ = 0.0119; Q = 1.01, df = 1, *p* = 0.315), indicating alignment between the two studies [33,35]. The non-significant effect, despite a trend toward risk reduction, demonstrates inconclusive protective benefits at this intermediate duration.

GLMM-based sensitivity analysis corroborates the primary results, indicating a borderline non-significant protective effect of biological augmentation against conversion to THA. Biological augmentation exhibited a non-significant trend toward reduced odds of clinical failure compared to control, with a pooled OR of 0.59 (95% CI: 0.34 to 1.05; Z = −1.81; *p* = 0.071). Heterogeneity was absent (τ^2^ = 0; τ = 0; I^2^ = 0.0%; H = 1.00), with non-significant tests for heterogeneity as assessed by the Wald test (Q = 0.00, df = 1, *p* = 1.00) and likelihood ratio test (LRT; Q = 2.59, df = 1, *p* = 0.108), suggesting complete consistency between the two included studies.

### 3.8. Summary of Results

Table 2 presents a summary of all key results from this meta-analysis. In this meta-analysis, the overall certainty of evidence ranged from low to moderate across the analyzed outcomes.

### 3.9. Risk of Bias Assessment

All included randomized trials demonstrated an overall low risk of bias according to the Cochrane RoB 2 assessment, supporting the reliability of the pooled estimates (Figure 18). Minor concerns were noted only in one study (Hu et al., 2018 [24]) regarding missing outcome data, but this is unlikely to have materially affected the overall conclusions. The generally low risk across methodological domains strengthens confidence in the observed effects of biologically augmented core decompression compared with standard or structurally reinforced approaches.

All included observational studies demonstrated high methodological quality, with total NOS scores of 8 out of 9 (Table 3). Each study showed clear inclusion and exclusion criteria, well-defined interventions, and sufficient follow-up periods. The main limitation across studies was the absence of randomization and limited statistical control for confounding variables, which may introduce selection bias. Overall, the evidence can be considered reliable for observational research, but results should be interpreted with caution regarding causal inference.

## 4. Discussion

The present meta-analysis provides a comprehensive synthesis of the comparative effectiveness of biological and structural augmentation strategies following core decompression in early-stage osteonecrosis of the femoral head. Across fourteen studies and over one thousand treated hips, the findings delineate distinct therapeutic trajectories between biologically driven and mechanically oriented interventions. Biological augmentation through agents such as platelet-rich plasma, bone marrow aspirate concentrate, and mesenchymal stem cell–based therapies was consistently associated with meaningful short- to mid-term functional and symptomatic improvement. These gains emerged rapidly within the first six months, stabilized by two years, and persisted across most studies with low to moderate heterogeneity. This pattern reinforces the concept that biological interventions primarily accelerate endogenous repair and angiogenic mechanisms rather than provide long-term mechanical reinforcement.

In contrast, structural augmentation techniques, including bone grafting and fibular support, failed to demonstrate significant functional or radiographic superiority over standard core decompression. The lack of durable benefit and high variability across studies suggest that structural reinforcement alone may not effectively counteract the multifactorial pathophysiology of osteonecrosis, which is dominated by ischemic and cellular failure rather than gross mechanical instability in early stages. The differential temporal patterns observed, rapid functional gains with biological augmentation versus plateauing or a decline with structural support, underscore a mechanistic divergence that holds clinical relevance for treatment selection.

These findings emphasize that early biological modulation of bone regeneration may offer the most promising treatment for hip preservation before subchondral collapse. By integrating angiogenic, osteogenic, and anti-inflammatory effects, biologically augmented decompression appears to optimize the early reparative window of osteonecrosis, potentially delaying radiological progression and the need for THA.

From a clinical standpoint, these findings suggest that biological augmentation should be prioritized in younger patients (typically under 40 years of age) and in early disease stages (ARCO I–II), where regenerative potential and bone remodeling capacity remain high. In these cases, biologically enhanced decompression may delay disease progression. Conversely, structural augmentation alone appears less effective for early-stage lesions and may be reserved for patients with more extensive necrotic areas or those with partial subchondral compromise where mechanical support is required. In daily orthopedic practice, careful patient selection based on age, stage, and lesion size may therefore optimize outcomes and minimize unnecessary surgical burden.

These findings are consistent with prior meta-analysis [38] demonstrating the overall safety and effectiveness of CD in early osteonecrosis of the femoral head. In a large-scale review including 32 studies and 1865 patients (2441 hips), the pooled success rate of CD was approximately 65%, confirming its role as an effective joint-preserving strategy, particularly in early-stage disease. The authors reported that combining CD with autologous bone or bone marrow significantly increased success rates and reduced the risk of radiographic progression and conversion to total hip arthroplasty, whereas isolated CD showed lower durability across stages. Importantly, outcomes varied by disease stage, with diminished benefits observed in advanced osteonecrosis, underscoring the critical timing of intervention. These results align with the present meta-analysis, in which biologically augmented decompression yielded greater and more sustained functional improvements than structural approaches, further supporting the role of biologically enhanced regeneration rather than mechanical reinforcement as the key determinant of long-term joint preservation.

These results are further supported by recent evidence from Tang et al. [7], who conducted a systematic review and meta-analysis of seven randomized controlled trials and eight cohort studies involving 954 hips. The authors demonstrated that core decompression combined with regenerative therapy significantly reduced the risk of disease progression and conversion to total hip arthroplasty compared with CD alone (pooled RR ≈ 0.55), with the most pronounced benefit observed in the bone marrow aspirate concentrate subgroup (stage progression RR = 0.47; THA conversion RR = 0.41). Functional outcomes, including HHS, VAS, WOMAC, and Lequesne scores, were consistently improved with regenerative augmentation, particularly among younger patients (<40 years) and those receiving low-dose BMAC. These findings align closely with the present analysis, reinforcing the superior functional and symptomatic profile of biologically augmented decompression relative to structural reinforcement. Collectively, both meta-analyses highlight that biologically driven repair—via cell-based or growth factor–mediated mechanisms—offers not only early symptomatic relief but also a plausible delay in femoral head collapse in appropriately selected early-stage cases.

The pooled MDs in HHS observed for biological augmentation, while statistically significant at multiple time points, must be evaluated against clinically meaningful benchmarks to guide practical application. For instance, the MD of 8.39 (95% CI: 5.63 to 11.15) at 6 months exceeds commonly reported MCID thresholds for HHS in hip preservation contexts (typically 8–10 points), indicating substantial early functional and pain-related benefits that patients are likely to perceive as meaningful [39,40]. However, the attenuated MD of 3.65 (95% CI: 0.22 to 7.08) at 12 months, despite achieving statistical significance (*p* = 0.037), falls below these thresholds, signalizing limited clinical relevance at this interval and a potential plateau in therapeutic effects. By 24 months, the MD of 5.04 (95% CI: 2.26 to 7.81) approaches but does not consistently surpass the MCID, underscoring sustained yet moderated durability. In contrast, structural augmentation yielded non-significant and often negative MDs (e.g., −2.53 at 12 months), which are well below any MCID and imply negligible or potentially detrimental clinical impact. These interpretations highlight the importance of prioritizing interventions where effects exceed MCID values, particularly in early-stage osteonecrosis management, while advocating for patient-centered outcomes in future trials to refine these benchmarks.

The present meta-analysis is subject to several limitations that warrant consideration when interpreting its findings. First, the small number of included studies per analysis (ranging from 2 to 4) and modest sample sizes (total participants or hips often below 200 per time point) constrained statistical power, potentially contributing to wide confidence intervals and non-significant results in some subgroups, as is common in meta-analyses with limited data [17]. This scarcity also precluded formal assessments of publication bias, such as Egger’s test, due to insufficient studies (typically requiring ≥10 for reliable detection) [41], raising the possibility of selective reporting favoring positive outcomes.

Second, substantial heterogeneity was observed in several analyses (e.g., I^2^ > 70% for HHS at 12 months in biological augmentation and VAS at 6 months in growth-factor/platelet-based approaches), likely attributable to variations in patient demographics (e.g., age, sex distribution, and disease staging across ARCO/Ficat I-II), intervention specifics (e.g., cell sourcing CD protocols, MSC origin, PRP dosage, and imaging assessment methods), and study designs (mix of RCTs and retrospective studies), which may introduce confounding and limit generalizability [23]. Although random-effects models and subgroup stratifications were employed to mitigate this, the underlying clinical diversity—predominantly from Chinese cohorts (6 of 14 studies)—may not fully represent global populations, potentially biasing estimates toward region-specific practices [14]. Furthermore, variability in MCID estimates across populations (e.g., due to differences in disease staging or demographics) may confound the clinical interpretability of HHS effects, warranting standardized reporting in primary studies [31].

Third, the reliance on short- to medium-term follow-up (up to 60 months) restricted insights into long-term durability, particularly for binary outcomes like radiological progression and THA conversion, where progression may manifest beyond assessed intervals. Sparse event data in binary analyses necessitated continuity corrections, which can introduce bias in low-event scenarios, though REML estimation for τ^2^ aimed to enhance robustness [21]. Indirect group comparisons were limited to shared time points (e.g., 12 and 24 months for HHS), assuming control arm comparability, but lacked power for definitive conclusions on relative superiority.

A further limitation is the reliance on hip-level rather than patient-level data, which may introduce clustering bias in patients with bilateral ONFH, potentially underestimating variance and leading to overly precise estimates. Although bilateral involvement is common in ONFH (affecting 20–50% of cases), its prevalence in early-stage disease is typically low and balanced across study arms, minimizing substantial bias. Nonetheless, access to individual patient data in future meta-analyses would enable clustered analyses to refine precision. This issue is inherent to aggregated data syntheses and does not appear to have altered our overall conclusions, as sensitivity analyses showed consistent results. Additionally, exclusion of single-study data adhered to methodological rigor but may have omitted valuable exploratory insights, while the absence of data for certain subgroups and time points curtailed comprehensive evaluations [16]. Sensitivity analyses were performed, yet the overall risk of bias from included studies could not be fully eliminated, potentially inflating effect estimates.

Future research should incorporate patient-reported MCID assessments to better delineate clinically relevant thresholds, ensuring that meta-analyses can more robustly inform evidence-based practice.

These constraints highlight the preliminary nature of the evidence and underscore the need for larger, multicenter RCTs with standardized protocols, extended follow-up, and diverse populations to validate and refine these findings.

## 5. Conclusions

Biological augmentation of core decompression provides consistent functional and symptomatic benefits in early-stage osteonecrosis of the femoral head, with improvements in HHS and pain reduction maintained up to two years. In contrast, structural augmentation offers no significant functional advantage and shows greater variability across studies. The results support the use of biologically enhanced decompression as a first-line strategy for hip preservation prior to subchondral collapse. Large-scale, standardized randomized trials are warranted to confirm these findings and establish long-term outcomes. Further research should focus on the standardization of biologically augmented CD protocols, particularly regarding cell quantity and delivery methods.

## Figures and Tables

**Figure 1 medsci-13-00258-f001:**
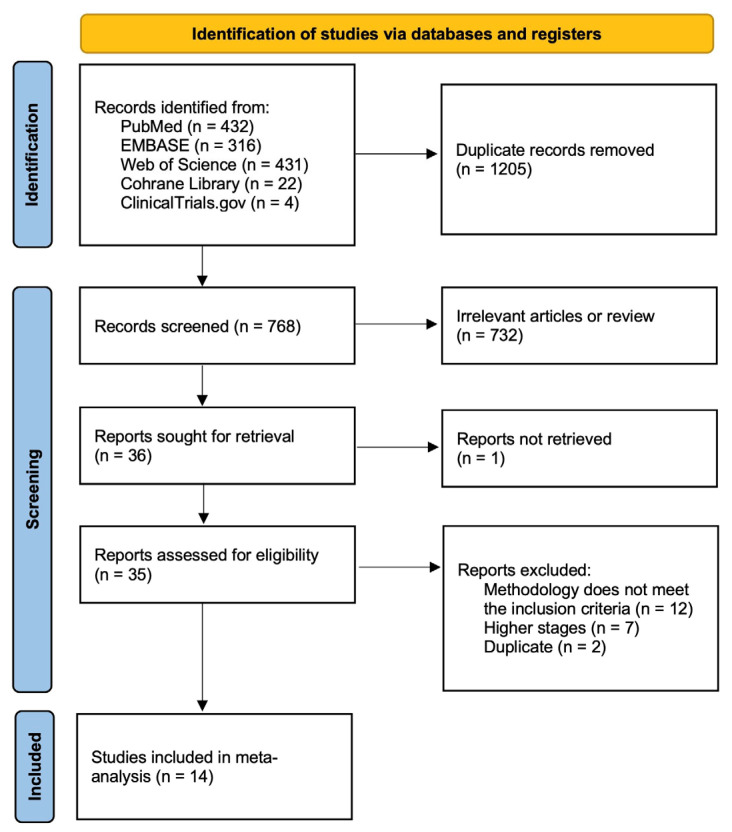
Flow diagram of literature search and selection for meta-analysis [14].

**Figure 2 medsci-13-00258-f002:**
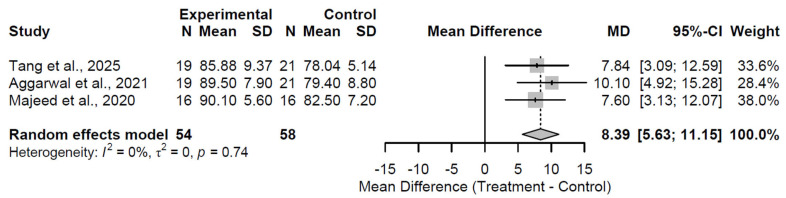
Forest plot of pooled mean differences in Harris Hip Score for biological augmentation versus control at 6 months follow-up [27,32,36].

**Figure 3 medsci-13-00258-f003:**
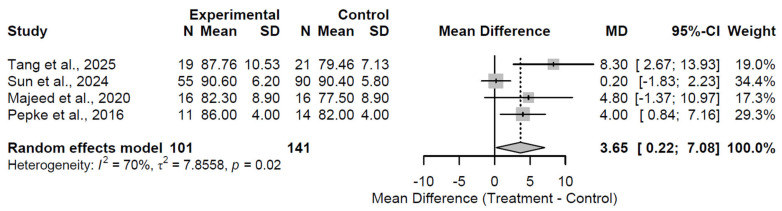
Forest plot of pooled mean differences in Harris Hip Score for biological augmentation versus control at 12 months follow-up [27,31,32,37].

**Figure 4 medsci-13-00258-f004:**
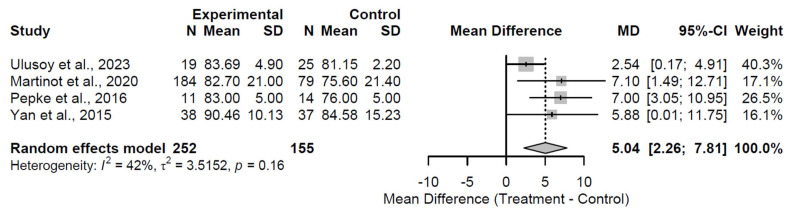
Forest plot of pooled mean differences in Harris Hip Score for biological augmentation versus control at 24 months follow-up [28,30,33,37].

**Figure 16 medsci-13-00258-f016:**
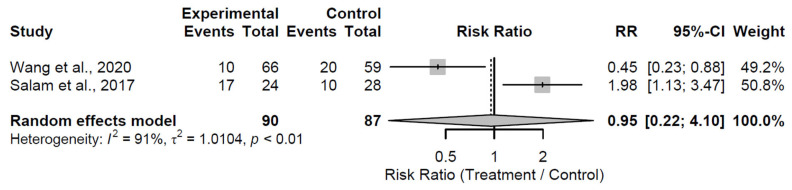
Forest plot of pooled risk ratios for radiological progression in structural augmentation versus control at 36 months follow-up [25,26].

**Figure 17 medsci-13-00258-f017:**
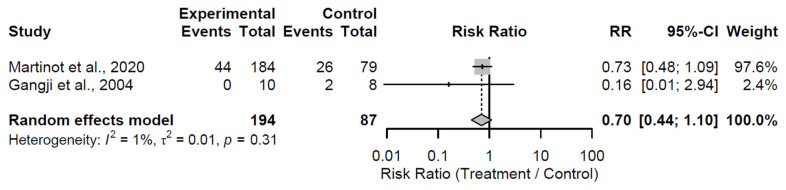
Forest plot of pooled risk ratios for conversion to Total Hip Arthroplasty in biological augmentation versus control at 24 months follow-up [33,35].

**Figure 18 medsci-13-00258-f018:**
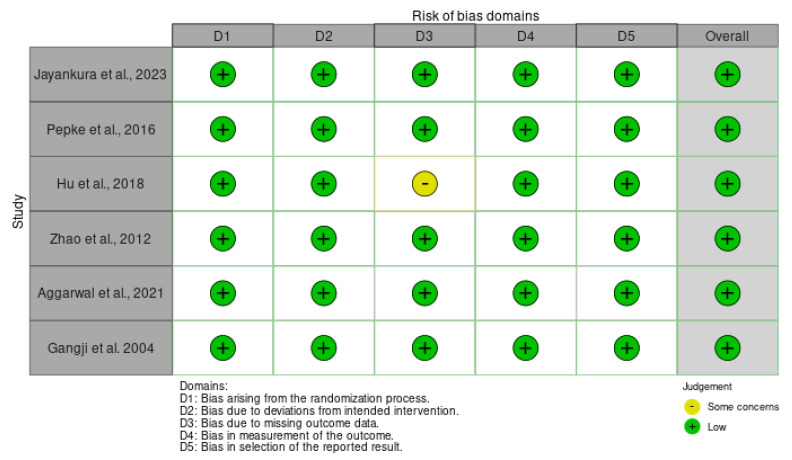
Risk of bias assessment for randomized controlled trials included in the meta-analysis [24,29,34,35,36,37].

**Table 1 medsci-13-00258-t001:** Characteristics of included studies.

Study	Country	Study Type	Intervention	Comparator (Control)	Mean Age (Treatment)	Mean Age (Control)	Follow-Up (Months)	Stage
Hu et al., 2018 [24]	China	RCT	Fibula fixation	CD	40.38 ± 6.63	40.83 ± 6.73	48	Early
Wang et al., 2020 [25]	China	Retrospective	Light bulb	CD	38.1 ± 10.0	39.1 ± 10.2	48	ARCO II
Sallam et al., 2017 [26]	Egypt	Retrospective	Inverted femoral head graft	CD	32.7 ± 8.2	33.2 ± 8.8	36	ARCO I–II
Tang et al., 2025 [27]	China	Retrospective	CD + PRP	CD	46.58 ± 7.46	45.19 ± 10.61	11.6 vs. 11.3	ARCO I–II
Ulusoy et al., 2023 [28]	Turkey	Retrospective	CD + BMMSC	CD	38.4 ± 6.7	39.3 ± 6.5	31.8 vs. 32.2	Steinberg I–II
Jayankura et al., 2023 [29]	Belgium, the Netherlands, France, Germany, and the United Kingdom	RCT	CD + autologous osteoblastic cell transplantation	CD	46 ± 10	45 ± 10	17 vs. 18	ARCO I–II
Yan et al., 2015 [30]	China	Retrospective	CD + autologous MSC	CD	39.62 ± 11.83	37.24 ± 10.54	Mean 26	ARCO I–II
Sun et al., 2024 [31]	China	Retrospective	1. CD + AL 2. CD + BG 3. NVAF + BG 4. DAA + BG 5. PNHF + BG	CD	1. 39.5 ± 12.2 2. 39.3 ± 10.4 3. 36.5 ± 7.8 4. 38.1 ± 10.0 5. 38.2 ± 8.2	38.2 ± 8.2	60	ARCO II
Majeed et al., 2020 [32]	Iraq	Prospective comparative study	CD + PRP	CD	34.6		12	Ficat and Arlet I–II
Martinot et al., 2020 [33]	France	Retrospective	CD + biological augmentation	CD	ND	ND	24	Ficat I–II
Zhao et al., 2012 [34]	China	RCT	CD + BMMSC	CD	32.7 ± 10.5	33.8 ± 7.70	60	ARCO I–II
Gangji et al., 2004 [35]	Belgium	prospective cohort study	CD + Bone-Marrow-Graft	CD	40.9 ± 9.8	48.8 ± 11.2	24	ARCO I–II
Aggarwal et al., 2021 [36]	India	RCT	CD + PRP	CD	38.2 ± 10.4	35.2 ± 12.5	64.3 vs. 63.7	ARCO I–II
Pepke et al., 2016 [37]	Germany	Randomized Prospective Study	CD + BMAC	CD	44.3 ± 3.4	44.5 ± 3.3	24	ARCO II

ARCO—Association Research Circulation Osseous; BMMSC—bone marrow-derived mesenchymal stem cells; BMAC—bone marrow aspirate concentrate; CD—core decompression; CD  +  AL—core decompression with oral alendronate; CD  +  BG—core decompression with bone grafting; DAA  +  BG—percutaneous femoral neck–head fenestration with bone grafting via direct anterior approach; NVAF  +  BG—nonvascularized allogeneic fibula with bone grafting; PNHF  +  BG—percutaneous femoral neck–head fenestration with bone grafting; PRP—platelet-rich plasma [24,25,26,27,28,29,30,31,32,33,34,35,36,37].

**Table 2 medsci-13-00258-t002:** Summary of all key results from this meta-analysis.

Outcome	Comparison	Follow-Up (Months)	Pooled Effect (95% CI)	Effect Direction	Heterogeneity (I^2^)	Certainty (GRADE)
HHS	Biological augmentation vs. control	6	MD = 8.39 (5.63–11.15), *p* < 0.001	Favors biological	0%	Moderate
		12	MD = 3.65 (0.22–7.08), *p* = 0.037	Favors biological	70%	Low
		24	MD = 5.04 (2.26–7.81), *p* < 0.001	Favors biological	42%	Moderate
HHS	Structural augmentation vs. control	12	MD = −2.53 (−7.67 to 2.62), *p* = 0.336	NS	98%	Low
		24	MD = −2.29 (−13.34 to 8.75), *p* = 0.684	NS	99%	Low
		48	MD = −2.73 (−13.67 to 8.22), *p* = 0.625	NS	99%	Low
VAS	Biological augmentation vs. control	3	MD = −0.23 (−0.58 to 0.12), *p* = 0.198	Favors biological	0%	Low
		6	MD = −0.86 (−1.39 to −0.33), *p* = 0.002	Favors biological	72%	Moderate
		12	MD = −0.66 (−0.89 to −0.43), *p* < 0.001	Favors biological	0%	Moderate
		24	MD = −0.50 (−0.75 to −0.25), *p* < 0.001	Favors biological	0%	Moderate
Radiological progression	Biological augmentation vs. control	12	RR = 0.58 (0.28–1.19), *p* = 0.137	NS	12%	Low
		60	RR = 0.37 (0.14–1.02), *p* = 0.054	NS	55%	Low
Radiological progression	Structural augmentation vs. control	36	RR = 0.95 (0.22–4.10), *p* = 0.949	NS	91%	Low
Conversion to THA	Biological augmentation vs. control	24	RR = 0.70 (0.44–1.10), *p* = 0.124	NS	1%	Low

MD—mean difference; RR—risk ratio; THA—total hip arthroplasty; HHS—Harris Hip Score; VAS—Visual Analog Scale.

**Table 3 medsci-13-00258-t003:** Quality assessment of the included observational studies using the Newcastle–Ottawa Scale (NOS).

Study	Selection (Max 4)	Comparability (Max 2)	Outcome (Max 3)	Total (Max 9)
Wang et al., 2020 [25]	4	1	3	8
Sallam et al., 2017 [26]	4	1	3	8
Tang et al., 2025 [27]	4	1	3	8
Ulusoy et al., 2023 [28]	4	1	3	8
Yan et al., 2015 [30]	4	1	3	8
Sun et al., 2024 [31]	4	1	3	8
Majeed et al., 2020 [32]	4	1	3	8
Martinot et al., 2020 [33]	4	1	3	8

## Data Availability

The original data presented in the study are openly available in FigShare: Konarski, Wojciech (2025). Effectiveness of Different Types of Core Decompression in Early-Stage Osteonecrosis of the Femoral Head: A Systematic Review and Meta-Analysis-Raw Data. Figshare at https://doi.org/10.6084/m9.figshare.30327922.v1.

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
