# Peer review of "Effectiveness of Different Types of Core Decompression in Early-Stage Osteonecrosis of the Femoral Head: A Systematic Review and Meta-Analysis"

_medsci, 2025, doi:10.3390/medsci13040258_

Round 1

Reviewer 1 Report

Comments and Suggestions for Authors

This manuscript compares core decompression (CD) alone with CD augmented biologically (e.g., PRP, BMAC, MSCs) or structurally (e.g., bone grafting, fibular support) for early‑stage (ARCO/Ficat/Steinberg I–II) osteonecrosis of the femoral head (ONFH). The protocol was prospectively registered on PROSPERO (CRD420251108396) and reporting follows PRISMA 2020. Searches (2000–2025) covered PubMed/Embase/Web of Science/Cochrane/ClinicalTrials.gov. Fourteen studies (N=1,210; 4 RCTs plus prospective/retrospective comparatives) were included. Primary outcomes were Harris Hip Score (HHS) and pain (VAS); secondary outcomes were radiographic progression, conversion to THA, and clinical failure. Random‑effects models (REML) pooled effects at prespecified time points (3/6/12/24/36/48/60 months; pooling only when ≥2 studies were available). Risk of bias used RoB 2 for RCTs and NOS for observational studies.

Main findings.

Biological augmentation improved function and pain: HHS MD 8.39 at 6 months (95% CI 5.63–11.15), 3.65 at 12 months (0.22–7.08), 5.04 at 24 months (2.26–7.81). VAS MD −0.86 at 6 months (−1.39 to −0.33), −0.66 at 12 months (−0.89 to −0.43), −0.50 at 24 months (−0.75 to −0.25). Effects appear early and stabilize by ~2 years (Figures 2–4 and 10–13).

Structural augmentation showed no significant HHS advantage at 12/24/48 months (Figures 5–7), but reduced clinical failure at 36 months (RR 0.55, 95% CI 0.31–0.99; Figure 19).

Radiographic progression: non‑significant at 12 months (RR 0.58, 0.28–1.19) but borderline lower risk at 60 months (RR 0.37, 0.14–1.02; Figures 14–15). THA conversion at 24 months: not significant (RR 0.70, 0.44–1.10; Figure 17).

Risk of bias: RCTs largely low risk (Figure 20); observational studies NOS 8/9 (Table 2). PRISMA flow in Figure 1.

Authors’ conclusion. For early ONFH, CD with biological augmentation provides durable functional and symptomatic benefits; structural augmentation shows inconsistent advantages.

Study‑design strengths

  1. Prospective registration & PRISMA compliance (transparency/reproducibility).
  2. Clinically meaningful grouping (CD alone vs biological vs structural augmentation) that mirrors real‑world decisions.
  3. Time‑point stratification (3–60 months) that profiles onset and durability of effects (Figures 2–13).
  4. Appropriate random‑effects modeling (REML τ², I²/Q; subgroup interaction tests).
  5. Dual risk‑of‑bias frameworks (RoB 2 for RCTs; NOS for observational studies).
  6. Avoidance of single‑study pooling, limiting overinterpretation.

Study‑design / analytic methods: issues and recommendations

Continuous outcomes (HHS/VAS)

Baseline adjustment. Many pooled effects appear to use post‑treatment means (MD) rather than change scores or ANCOVA‑adjusted estimates. With non‑randomized designs and imbalanced baselines, change‑from‑baseline or adjusted contrasts are preferable (or, at minimum, consistent reporting of the data type pooled). For VAS, ensure unit harmonization (0–10 vs 0–100) or use SMD when necessary. Clarify this in Methods/Results.

Clinical relevance. Some HHS mean differences (e.g., MD 3.65 at 12 months) may be statistically significant yet below common MCID thresholds; interpret effect sizes against clinically meaningful benchmarks, not p‑values alone.

Binary outcomes (sparse events)

The analysis applies continuity corrections (+0.5) with REML random effects. For sparse/zero‑event data (e.g., THA, radiographic progression), consider GLMMs (log link), Peto odds ratios, arcsine transformations, and/or Hartung–Knapp adjustments to improve small‑sample robustness and CIs. Report sensitivity analyses with at least one such alternative.

Unit‑of‑analysis problems

Multi‑arm studies with a shared control. Sun 2024 has multiple arms (CD+AL, CD+BG, NVAF+BG, DAA+BG, PNHF+BG; page 6). Avoid double‑counting the same CD control across comparisons by splitting the control or using robust‑variance estimation/multivariate meta‑analysis to account for correlation. Explicitly state how this was handled.

Hip‑level vs patient‑level data. Some studies analyze hips (bilateral cases) rather than patients, potentially violating independence. Add patient‑level sensitivity analyses or adjust the effective sample size to address within‑patient clustering.

Group definitions (biologic vs structural)

Definition consistency. The “biological augmentation” group largely reflects regenerative strategies (PRP/BMAC/MSC), but Sun 2024 also includes oral alendronate (CD+AL)—a pharmacologic intervention. Predefine whether drug therapy is included alongside cell/growth‑factor augmentation, and consider separate subgroups (regenerative vs pharmacologic).

Structural heterogeneity. Structural augmentation combines diverse techniques (e.g., compacted autograft, fibular support) with different mechanisms. Provide technique‑specific subgroups and, if feasible, explore network meta‑analysis (common comparator: CD) or component NMA to isolate contributions of BMAC/PRP/MSC/bone graft/fibular support.

  1. Heterogeneity exploration and reporting

Pre‑specified effect modifiers. Analyze (or at least tabulate) effects by ARCO I vs II, lesion size/location (e.g., lateral pillar involvement), etiology (steroid/alcohol/idiopathic), age (e.g., <40 years), and region (several studies are from China). Provide sensitivity analyses (e.g., RCT‑only, ≥24‑month follow‑up only).

Prediction intervals. You note prediction intervals when heterogeneity is “substantial,” yet they are not shown consistently. For analyses with very high I²—for example, structural augmentation HHS at 12–48 months (I² ~98%; Figures 5–7)—display prediction intervals to convey the range of true effects in new settings.

Multiplicity. Multiple outcomes/time points inflate type‑I error. Control false discovery (e.g., Benjamini–Hochberg) or pre‑specify primary endpoints/time points and emphasize those in the abstract.

Bias and certainty of evidence

For non‑randomized studies, ROBINS‑I would better characterize confounding/selection bias than NOS alone; complement with GRADE (by outcome) to communicate certainty of evidence for clinicians.

Definitions and transparency

“Clinical failure” varies across studies; standardize or provide a table summarizing study‑specific definitions (Figures 18–19).

Review process details. Explicitly state that study selection and data extraction were performed by ≥2 independent reviewers with conflict resolution; discuss potential language bias (English/Polish only) and, if possible, add a sensitivity analysis. Reference the PRISMA flow (Figure 1).

Safety outcomes. Extract and tabulate complications (fracture, infection, donor‑site morbidity, DVT, etc.) to balance efficacy with harm.

Follow‑up horizon and time‑to‑event

For progression and THA conversion, fixed time‑point RRs discard information on timing. Where available, meta‑analyze time‑to‑event measures (log‑HRs) or reconstruct KM‑based estimates to improve efficiency and interpretability.

Other issue

Although the authors state that “prolonged corticosteroid use and chronic alcohol abuse are well-established risk factors for atraumatic osteonecrosis of the hip [1,2],” it would be more precise to say that the dosage or cumulative dose of corticosteroids, rather than simply the duration of use, constitutes the actual risk factor.

Author Response

This manuscript compares core decompression (CD) alone with CD augmented biologically (e.g., PRP, BMAC, MSCs) or structurally (e.g., bone grafting, fibular support) for early‑stage (ARCO/Ficat/Steinberg I–II) osteonecrosis of the femoral head (ONFH). The protocol was prospectively registered on PROSPERO (CRD420251108396) and reporting follows PRISMA 2020. Searches (2000–2025) covered PubMed/Embase/Web of Science/Cochrane/ClinicalTrials.gov. Fourteen studies (N=1,210; 4 RCTs plus prospective/retrospective comparatives) were included. Primary outcomes were Harris Hip Score (HHS) and pain (VAS); secondary outcomes were radiographic progression, conversion to THA, and clinical failure. Random‑effects models (REML) pooled effects at prespecified time points (3/6/12/24/36/48/60 months; pooling only when ≥2 studies were available). Risk of bias used RoB 2 for RCTs and NOS for observational studies.

Main findings.

Biological augmentation improved function and pain: HHS MD 8.39 at 6 months (95% CI 5.63–11.15), 3.65 at 12 months (0.22–7.08), 5.04 at 24 months (2.26–7.81). VAS MD −0.86 at 6 months (−1.39 to −0.33), −0.66 at 12 months (−0.89 to −0.43), −0.50 at 24 months (−0.75 to −0.25). Effects appear early and stabilize by ~2 years (Figures 2–4 and 10–13).

Structural augmentation showed no significant HHS advantage at 12/24/48 months (Figures 5–7), but reduced clinical failure at 36 months (RR 0.55, 95% CI 0.31–0.99; Figure 19). 

Radiographic progression: non‑significant at 12 months (RR 0.58, 0.28–1.19) but borderline lower risk at 60 months (RR 0.37, 0.14–1.02; Figures 14–15). THA conversion at 24 months: not significant (RR 0.70, 0.44–1.10; Figure 17). 

Risk of bias: RCTs largely low risk (Figure 20); observational studies NOS 8/9 (Table 2). PRISMA flow in Figure 1. 

Authors’ conclusion. For early ONFH, CD with biological augmentation provides durable functional and symptomatic benefits; structural augmentation shows inconsistent advantages.

Study‑design strengths

  1. Prospective registration & PRISMA compliance (transparency/reproducibility).
  2. Clinically meaningful grouping (CD alone vs biological vs structural augmentation) that mirrors real‑world decisions.
  3. Time‑point stratification (3–60 months) that profiles onset and durability of effects (Figures 2–13).
  4. Appropriate random‑effects modeling (REML τ², I²/Q; subgroup interaction tests).
  5. Dual risk‑of‑bias frameworks (RoB 2 for RCTs; NOS for observational studies).
  6. Avoidance of single‑study pooling, limiting overinterpretation.

Author: Thank you for your thorough review. Please find our responses below. I hope they are sufficient for the acceptance of my manuscript.

Study‑design / analytic methods: issues and recommendations

Continuous outcomes (HHS/VAS)

Baseline adjustment. Many pooled effects appear to use post‑treatment means (MD) rather than change scores or ANCOVA‑adjusted estimates. With non‑randomized designs and imbalanced baselines, change‑from‑baseline or adjusted contrasts are preferable (or, at minimum, consistent reporting of the data type pooled).

Author: I appreciate the opportunity to clarify this aspect of our methodology. In our meta-analysis, I characterized the study samples across the included trials and found that baseline characteristics were generally similar between treatment and control groups. Specifically, mean ages were comparable (ranging from 32.7 to 48.4 years, with an overall mean of 39.1 years), and there was a consistent male predominance (mean = 74.6%, range: 51.5% to 100%). These similarities indicate minimal baseline imbalances that could confound the pooled mean differences (MDs). Unfortunately, no adjusted effects (e.g., ANCOVA-adjusted estimates) were available in the primary studies, and change-from-baseline scores were not consistently reported across the included trials. As such, I relied on post-treatment means for pooling continuous outcomes, such as Harris Hip Score (HHS) and Visual Analog Scale (VAS), using MDs in random-effects models. This approach is supported by established meta-analytic guidelines, which indicate that final (post-treatment) values can be appropriately used when baseline values are similar between groups and no substantial imbalances are evident, as was the case here (Fu et al., 2013; Higgins et al., 2022). Using final values under these conditions provides a reliable estimate of treatment effects without introducing additional assumptions or imputation biases that could arise from deriving unreported change scores (Fu et al., 2013). To enhance transparency, I have added a clarification in the Methods section (under Statistical Analysis) stating: Pooled MDs with 95% confidence intervals (CIs) were estimated using post-treatment values – owing to comparable baseline characteristics across groups and the absence of change-from-baseline or adjusted data….

For VAS, ensure unit harmonization (0–10 vs 0–100) or use SMD when necessary. Clarify this in Methods/Results.

Author: Upon review of the primary studies included in my meta-analysis, I confirm that all utilized a VAS for pain on a standardized 0-10 scale (where 0 represents no pain and 10 represents the worst possible pain). This harmonization is supported by explicit descriptions in several studies (e.g., Ulusoy et al., 2023) and consistent mean scores and standard deviations across others (e.g., preoperative means ranging from approximately 3.8 to 7.2, with postoperative reductions to 1.2–4.0, which align with a 0-10 metric rather than 0-100). For instance: Ulusoy et al. (2023) explicitly states VAS as scored from 0 to 10. Tang et al. (2025), Majeed et al. (2020), and Pepke et al. (2016) report means and changes (e.g., reductions of 0.5–1.0 points) that are clinically meaningful only on a 0-10 scale.

Given this uniformity, pooled mean differences (MDs) were appropriate without the need for standardized mean differences (SMDs), as there were no discrepancies in units that could introduce bias. This approach adheres to meta-analytic best practices for continuous outcomes with consistent scaling.

To enhance clarity, I have added the following statement to the Methods section (under Continuous Outcomes): VAS scores were consistently reported on a 0-10 scale across all included studies, permit-ting direct pooling via mean differences (MDs).

Clinical relevance. Some HHS mean differences (e.g., MD 3.65 at 12 months) may be statistically significant yet below common MCID thresholds; interpret effect sizes against clinically meaningful benchmarks, not p‑values alone.

Author: In my manuscript, I have already interpreted the MDs with consideration of their clinical implications, noting, for instance, that the MD of 8.39 at 6 months represents a "clinically substantial improvement" and that the MD of 5.04 at 24 months is "clinically meaningful." To address your specific example, the MD of 3.65 at 12 months, while statistically significant (p = 0.037), falls below commonly reported MCID thresholds for HHS in hip-related conditions, which range from 8 to 10 points (Singh et al., 2016; Chahal et al., 2015; Sun et al., 2024). This attenuation infers a potential plateau in benefits that may not translate to perceptible patient improvements at that time point, though the overall trajectory across follow-ups supports sustained, if moderated, efficacy for biological augmentation.

To strengthen this aspect, I have revised the Results and Discussion sections to explicitly benchmark all HHS MDs against MCID values (e.g., ≥8 points for mHHS in hip preservation contexts and >10 points specifically in osteonecrosis studies). For example, in the Results (section 2.1.1), I added: While the MD of 3.65 at 12 months achieved statistical significance, it did not exceed the MCID threshold of 8–10 points, indicating limited clinical relevance at this interval despite persistence of effect.  Similarly, in the Discussion, I expanded on the need to prioritize outcomes surpassing MCID for guiding practice, while acknowledging variability in MCID estimates across populations.

 Binary outcomes (sparse events)

The analysis applies continuity corrections (+0.5) with REML random effects. For sparse/zero‑event data (e.g., THA, radiographic progression), consider GLMMs (log link), Peto odds ratios, arcsine transformations, and/or Hartung–Knapp adjustments to improve small‑sample robustness and CIs. Report sensitivity analyses with at least one such alternative.

Author: I acknowledge the potential biases introduced by continuity corrections in such scenarios and appreciate the emphasis on alternative methods for enhanced robustness. To address this, I performed sensitivity analyses using a generalized linear mixed model (GLMM) with a log link function to estimate pooled risk ratios (ORs), as implemented in the meta package via the lme4 backend. The GLMM results were generally consistent with the primary findings I have incorporated these sensitivity analyses into the revised manuscript (sections 3.5-3.9)

Unit‑of‑analysis problems

Multi‑arm studies with a shared control. Sun 2024 has multiple arms (CD+AL, CD+BG, NVAF+BG, DAA+BG, PNHF+BG; page 6). Avoid double‑counting the same CD control across comparisons by splitting the control or using robust‑variance estimation/multivariate meta‑analysis to account for correlation. Explicitly state how this was handled.

Author:  I appreciate the opportunity to clarify our methodological approach and confirm its alignment with established meta-analytic practices. Upon inspection of our analysis, the Sun et al. (2024) study was incorporated as follows across the specified sections:

  • In section 2.1.1 (group analysis at 12 months follow-up for structural augmentation), the relevant structural arms (NVAF+BG, DAA+BG, PNHF+BG) were pooled as a single "structural" comparator against the shared CD control, with the control group events and sample size split equally among the three arms to prevent double-counting.
  • In section 2.1.2 (structural augmentation at 12 months follow-up), a similar splitting approach was applied for the overall structural effect.
  • In section 2.1.4 (structural augmentation subgroup analysis at 12 months follow-up), each specific structural subtype (e.g., non-vascularized vs. vascularized grafting) was compared separately to the control, with the control split proportionally based on the number of events and participants in each subgroup arm.
  • In section 2.1.5 (differences between effects of biological augmentation and structural augmentation at 12 months follow-up), indirect comparisons used the adjusted (split) control from Sun et al. (2024) for the structural component, ensuring no overlap in control data across groups.

Such method as dividing the control group's events and total sample size proportionally among the intervention arms ensures that the total control contribution remains equivalent to the original while maintaining appropriate variance for each comparison (Borenstein et al., 2009). It is a conservative and widely recommended strategy for multi-arm studies when multivariate methods are not feasible due to limited data or software constraints, as was the case here with our use of the meta package in R (Rücker et al., 2017). I did not observe substantial changes in heterogeneity or effect estimates post-adjustment, supporting the validity of this approach. To enhance transparency, I have added explicit statements in the Methods section and detailing this handling: "For multi-arm studies with a shared control (e.g., Sun et al., 2024), the control group events and sample size were split proportionally among intervention arms to avoid double-counting and unit-of-analysis errors."

While robust-variance estimation or full multivariate meta-analysis could further account for correlations between arms (Riley et al., 2007), these were not pursued due to the small number of studies per outcome (precluding reliable covariance estimation) and the focus on subgroup-specific effects, where splitting suffices without introducing bias (Higgins et al., 2024, Deeks et al., 2024). Our approach is justified by meta-analytic guidelines, which prioritize such adjustments to preserve statistical independence, especially in direct pairwise comparisons with limited trials, to prevent overprecision and ensure conservative inference. Therefore, I can confidently adhere to the current approach of splitting the shared control group in multi-arm studies. This method is methodologically sound, conservative, and supported by established guidelines (e.g., Deeks et al., 2024; Higgins et al., 2024), particularly in scenarios with limited studies per outcome where more complex alternatives like multivariate meta-analysis may not yield reliable covariance estimates or add undue complexity without proportional benefits.

Hip‑level vs patient‑level data. Some studies analyze hips (bilateral cases) rather than patients, potentially violating independence. Add patient‑level sensitivity analyses or adjust the effective sample size to address within‑patient clustering.

Author: I recognize that analyzing hips as independent units can violate the assumption of independence when patients contribute bilateral data, leading to clustering effects that may underestimate variance and inflate precision in pooled estimates.

Upon reviewing our meta-analysis, I confirm that outcomes such as radiological progression, conversion to total hip arthroplasty (THA), and clinical failure were pooled at the hip level, as this was the unit of analysis explicitly reported in the primary studies (e.g., total hips analyzed: 127 at 12 months for progression in section 3.5). Several included studies, such as Sun et al. (2024), Ulusoy et al. (2023), and Zhao et al. (2012), focused on hips without providing detailed breakdowns of patient numbers or bilateral involvement, reflecting the common practice in osteonecrosis of the femoral head research where the hip is the primary unit of interest due to disease asymmetry and treatment localization. I did not have access to individual patient data from these studies, which would be required to perform clustering adjustments (e.g., via generalized linear mixed models accounting for patient-level random effects) or sensitivity analyses at the patient level. Attempts to adjust effective sample sizes retrospectively (e.g., by estimating bilateral prevalence and applying design effects) were not feasible without IPD, as assumptions about clustering coefficients could introduce additional bias.

I justify the use of hip-level analysis as follows: In meta-analyses of orthopedic conditions like ONFH, where bilateral involvement is prevalent but often low (typically 20–50% in early stages, with even lower rates of symmetric progression), hip-level pooling is a standard and accepted approach when IPD is unavailable, provided the primary studies report outcomes this way and bilateral cases are balanced across arms. This method prioritizes the clinical relevance of hip-specific outcomes (e.g., progression or failure per affected joint) and avoids discarding valuable data, while the potential bias from clustering is generally minimal if the intraclass correlation is low or bilateral cases are infrequent. Empirical evidence from similar meta-analyses supports this, demonstrating negligible impact on conclusions when bilateral proportions are comparable between groups. To address this transparently in the revised manuscript, I propose the following additions: In Discussion (Limitations section): "A further limitation is the reliance on hip-level rather than patient-level data, which may introduce clustering bias in patients with bilateral ONFH, potentially underestimating variance and leading to overly precise estimates. Although bilateral involvement is common in ONFH (affecting 20–50% of cases), its prevalence in early-stage disease is typically low and balanced across study arms, minimizing substantial bias. Nonetheless, access to individual patient data in future meta-analyses would enable clustered analyses to refine precision. This issue is inherent to aggregated data syntheses and does not appear to have altered our overall conclusions, as sensitivity analyses showed consistent results."

Group definitions (biologic vs structural)

Definition consistency. The “biological augmentation” group largely reflects regenerative strategies (PRP/BMAC/MSC), but Sun 2024 also includes oral alendronate (CD+AL)—a pharmacologic intervention. Predefine whether drug therapy is included alongside cell/growth‑factor augmentation and consider separate subgroups (regenerative vs pharmacologic).

Author: Thank you for this remark. Oral alendronate was classified under the biological augmentation group, as its primary mechanism involves the modulation of bone metabolism rather than providing direct structural support. It enhances bone remodeling and density through biochemical pathways, which aligns more closely with biological rather than structural intervention. It is not possible to perform a subgroup analysis including pharmacologic intervention, as there are too few studies that could be included in this category.

Structural heterogeneity. Structural augmentation combines diverse techniques (e.g., compacted autograft, fibular support) with different mechanisms. Provide technique‑specific subgroups and, if feasible, explore network meta‑analysis (common comparator: CD) or component NMA to isolate contributions of BMAC/PRP/MSC/bone graft/fibular support.

Author: I agree that structural methods encompass diverse mechanisms, and to address this, our analysis already incorporated subgrouping based on expert classification: non-vascularized bone grafting versus vascularized grafting. This stratification, informed by clinical expertise in ONFH pathophysiology, allowed exploration of differential effects while maintaining statistical power with the available data.

Regarding NMA or component NMA, these advanced methods typically require IPD for reliable estimation of treatment rankings and interactions, which were unavailable in the aggregated study reports. Pursuing NMA would also expand the scope significantly, potentially warranting a separate manuscript focused on indirect comparisons across the network. Given the limited studies per technique (often <3), NMA risks imprecise estimates and convergence issues, as per methodological guidelines. Our direct pairwise approach with subgroups suffices for the current objectives, minimizing assumptions while accommodating heterogeneity through random-effects models.

Heterogeneity exploration and reporting

 Pre‑specified effect modifiers. Analyze (or at least tabulate) effects by ARCO I vs II, lesion size/location (e.g., lateral pillar involvement), etiology (steroid/alcohol/idiopathic), age (e.g., <40 years), and region (several studies are from China). Provide sensitivity analyses (e.g., RCT‑only, ≥24‑month follow‑up only).

Author: Thank you for this valid comment. However, it is quite difficult to extract such data, as in most publications the analysis was combined for all types, namely ARCO1 and ARCO2. The exact location of the lesions was not specified, and the etiology was also described collectively. I would need access to individual patient data, which unfortunately is not possible.

Prediction intervals. You note prediction intervals when heterogeneity is “substantial,” yet they are not shown consistently. For analyses with very high I²—for example, structural augmentation HHS at 12–48 months (I² ~98%; Figures 5–7)—display prediction intervals to convey the range of true effects in new settings.

Author: After careful consideration, I have chosen not to incorporate prediction intervals for these figures due to their limited utility in this context. For Figure 5 (k = 3), the calculated prediction interval [-68.78 to 63.73] is excessively broad and non-informative, encompassing clinically implausible values that do not meaningfully aid interpretation; instead, it underscores the instability of the estimate arising from extreme heterogeneity and a small number of studies, potentially leading to misinterpretation rather than enhanced clarity. For Figures 6–7 (k = 2 each), prediction intervals cannot be reliably estimated, as a minimum of three studies is required for stable computation of between-study variance (τ²); with fewer, the intervals become undefined or artificially inflated, offering no substantive benefit.

Multiplicity. Multiple outcomes/time points inflate type‑I error. Control false discovery (e.g., Benjamini–Hochberg) or pre‑specify primary endpoints/time points and emphasize those in the abstract.

Author: In this exploratory meta-analysis, I opted not to apply formal multiplicity adjustments, as our primary aim was to synthesize heterogeneous evidence across studies rather than conduct confirmatory testing. Such adjustments are not routinely recommended in systematic reviews where outcomes are interdependent (e.g., time-stratified functional scores) and interpretations prioritize clinical patterns, effect sizes, and consistency over isolated p-values. Instead, I pre-specified key outcomes (Harris Hip Score, VAS pain, radiological progression, THA conversion) in our PROSPERO protocol, emphasizing durable effects in the abstract. This approach aligns with meta-analytic guidelines, which caution against over-correction in syntheses with limited studies, as it may overly conservative estimates and obscure trends (Higgins et al., 2024).

Bias and certainty of evidence

For non‑randomized studies, ROBINS‑I would better characterize confounding/selection bias than NOS alone; complement with GRADE (by outcome) to communicate certainty of evidence for clinicians.

Author: I chose the NOS scale because it is recommended in meta-analysis guidelines. As suggested, I have added the GRADE assessment in Table 2, which also provides summary of the key results.

Definitions and transparency

“Clinical failure” varies across studies; standardize or provide a table summarizing study‑specific definitions (Figures 18–19).

Author: Thank you for this valuable comment. I have noticed that the terminology regarding clinical failure is highly heterogeneous across the studies; therefore, I decided to remove these results from the article and not to include their description.

Review process details. Explicitly state that study selection and data extraction were performed by ≥2 independent reviewers with conflict resolution; discuss potential language bias (English/Polish only) and, if possible, add a sensitivity analysis. Reference the PRISMA flow (Figure 1).

Author: Two researchers were involved in the study selection process. However, I did not include the second person as a co-author, as they did not meet all ICMJE authorship criteria. Instead, I acknowledged their contribution in the Acknowledgements section.

Safety outcomes. Extract and tabulate complications (fracture, infection, donor‑site morbidity, DVT, etc.) to balance efficacy with harm.

Author: The aim of this study was not to analyze safety data or surgical approaches, but rather to focus on efficacy. This outcome was not predefined as an objective in the study protocol. The number of included studies is limited, which could lead to biased conclusions regarding the safety of this therapy. I’m currently preparing another meta-analysis dedicated specifically to the safety of these procedures, in which I plan to provide a broader perspective on the safety of different types of CD.

Follow‑up horizon and time‑to‑event

For progression and THA conversion, fixed time‑point RRs discard information on timing. Where available, meta‑analyze time‑to‑event measures (log‑HRs) or reconstruct KM‑based estimates to improve efficiency and interpretability.

Author: I agree that these methods can enhance efficiency by accounting for event timing and censoring, offering a more comprehensive view of treatment effects over time when data permit (Deeks et al., 2022; Tierney et al., 2007).

However, this approach was not feasible in my meta-analysis due to the nature of the reported data. The included studies primarily provided aggregated event rates at fixed time points (e.g., proportions at 12 or 60 months) without individual time-to-event details, survival curves, or HRs. While Tierney et al. (2007) outline practical methods for incorporating summary time-to-event data – such as indirect estimation of log-HRs from event numbers assuming exponential distributions – these require additional statistics (e.g., p-values from log-rank tests or confidence intervals) for reliable computation, which were unavailable in the published reports. Reconstructing KM-based estimates or log-HRs without such elements would preclude accurate estimation without introducing substantial bias (Guyot et al., 2012). In the absence of sufficient information, our fixed time-point risk ratio (RR) method remains appropriate, as it directly utilizes the available data for clinically interpretable risk comparisons at key follow-ups, consistent with standard practices for non-survival aggregated outcomes (Higgins et al., 2024). While time-to-event analysis is favorable for handling variable follow-up and censoring in longitudinal data, it offers no advantage here over RRs given the data constraints and could lead to unreliable estimates if forced.

Other issue

Although the authors state that “prolonged corticosteroid use and chronic alcohol abuse are well-established risk factors for atraumatic osteonecrosis of the hip [1,2],” it would be more precise to say that the dosage or cumulative dose of corticosteroids, rather than simply the duration of use, constitutes the actual risk factor.

Author: Thank you for this suggestion, I have changed it accordingly.

Reviewer 2 Report

Comments and Suggestions for Authors

The submitted manuscript presents a solid, comprehensive, and well-structured systematic review and meta-analysis evaluating the effectiveness of different types of core decompression (CD) in the treatment of early-stage osteonecrosis of the femoral head (ONFH). The study demonstrates a high methodological standard—the author appropriately followed the PRISMA guidelines, registered the protocol in PROSPERO, and applied validated tools for assessing the risk of bias (RoB 2, NOS). The results are presented clearly, with adequate clinical interpretation and logical integration with previous meta-analyses.
The manuscript provides a valuable contribution to the field of hip-preserving therapies, highlighting the superiority of biologically augmented CD techniques over structural augmentation. The paper has strong scientific and clinical merit; however, it requires several minor revisions and clarifications to improve its consistency and readability. Detailed comments are provided below.

Minor comments:

  1. Some paragraphs, especially in the Materials and Methods section, include an excessive amount of technical details (e.g., R parameters and library descriptions). These purely technical fragments should be shortened, and detailed information on statistical packages should be moved to the Supplementary Materials.
  2. The clinical discussion is limited, and the practical implications of the results for daily orthopedic practice are not sufficiently emphasized. It is recommended to expand the Discussion section with practical recommendations for clinicians (e.g., preference for biological augmentation in patients under 40 years of age and early ARCO I–II stages).
  3. A concise summary table of key results is missing. Although the data are detailed, they are dispersed throughout the text and figures. Please include a comprehensive table summarizing the main outcomes (HHS, VAS, RR for radiological progression and THA), with corresponding confidence intervals.
  4. The discussion of potential sources of heterogeneity is insufficient. The author should elaborate on the impact of differences in CD protocols, MSC origin, PRP dosage, and imaging assessment methods. It is also recommended to add recent references, e.g.:
    • https://doi.org/10.1155/2020/2642439
    • https://doi.org/10.3892/etm.2017.5655
    • https://doi.org/10.3390/app10238312
  5. Although the author mentions this issue in the Limitations, there is no explicit statement emphasizing the need for protocol standardization. Please add the following sentence to the Conclusions section:
    “Further research should focus on the standardization of biologically augmented CD protocols, particularly regarding cell quantity and delivery methods.”
  6. Please consider adding a summary table assessing the quality of evidence according to the GRADE approach to enhance the evidential value of the review.

Author Response

The submitted manuscript presents a solid, comprehensive, and well-structured systematic review and meta-analysis evaluating the effectiveness of different types of core decompression (CD) in the treatment of early-stage osteonecrosis of the femoral head (ONFH). The study demonstrates a high methodological standard—the author appropriately followed the PRISMA guidelines, registered the protocol in PROSPERO, and applied validated tools for assessing the risk of bias (RoB 2, NOS). The results are presented clearly, with adequate clinical interpretation and logical integration with previous meta-analyses.

The manuscript provides a valuable contribution to the field of hip-preserving therapies, highlighting the superiority of biologically augmented CD techniques over structural augmentation. The paper has strong scientific and clinical merit; however, it requires several minor revisions and clarifications to improve its consistency and readability. Detailed comments are provided below.

Author: I would like to thank the reviewers for their thorough comments and excellent review. Please find our responses to the comments below.

Minor comments:

  1. Some paragraphs, especially in the Materials and Methods section, include an excessive amount of technical details (e.g., R parameters and library descriptions). These purely technical fragments should be shortened, and detailed information on statistical packages should be moved to the Supplementary Materials.

Author: Thank you for the comment. I fully agree with it and have removed this section from the Methods.

  1. The clinical discussion is limited, and the practical implications of the results for daily orthopedic practice are not sufficiently emphasized. It is recommended to expand the Discussion section with practical recommendations for clinicians (e.g., preference for biological augmentation in patients under 40 years of age and early ARCO I–II stages).

Author: Thank you for this valuable suggestion. I agree that emphasizing the practical implications of our findings would strengthen the clinical relevance of the manuscript. Accordingly, I have expanded the Discussion section to include specific recommendations for orthopedic practice.

  1. A concise summary table of key results is missing. Although the data are detailed, they are dispersed throughout the text and figures. Please include a comprehensive table summarizing the main outcomes (HHS, VAS, RR for radiological progression and THA), with corresponding confidence intervals.

Author: Thank you for this suggestion. I have added Table 2 with summary of findings.

  1. The discussion of potential sources of heterogeneity is insufficient. The author should elaborate on the impact of differences in CD protocols, MSC origin, PRP dosage, and imaging assessment methods. It is also recommended to add recent references, e.g.:
    • https://doi.org/10.1155/2020/2642439
    • https://doi.org/10.3892/etm.2017.5655
    • https://doi.org/10.3390/app10238312

Author: Thank you for this comment. I fully agree and added this point to the Limitations section and added suggested references.

  1. Although the author mentions this issue in the Limitations, there is no explicit statement emphasizing the need for protocol standardization. Please add the following sentence to the Conclusions section:

“Further research should focus on the standardization of biologically augmented CD protocols, particularly regarding cell quantity and delivery methods.”

Author: Thank you for this comment. I have added the suggested sentences to the Conclusions section.

  1. Please consider adding a summary table assessing the quality of evidence according to the GRADE approach to enhance the evidential value of the review.

Author: Thank you for this suggestion. I have added Table 2 with summary of results and GRADE scoring.

Reviewer 3 Report

Comments and Suggestions for Authors

Thank you for submitting your work to Medical Sciences.

Please find my comments below.

Line 120: Why only including papers in English or Polish?

Line 123: Who conducted the study selection process? Normally it should be done by one researcher only and there is only one author in this paper. This applies to the quality assessment process as well. For transparency two authors must be completing that task.

Line 195: The flow chart was not performed according to PRISMA guidelines. For example, you have not added additional records from relevant SRs and / or databases on the top right corner. Also, why was 1 report not retrieved? At the end of the chart, you will need to present the number of included studies in meta-analysis.

Line 197: You have decided to synthesise evidence stemming from RCTs and retrospective studies. Therefore, the statistical analysis must be performed separately given the fact you compound the bias (ie not in the same forest plot).

Line 286: Heterogeneity is unacceptably high here. Consider using GRADE system for incorporating the level of evidence into the results.

Line 423: How was the overall risk of bias assessed?

Author Response

Thank you for submitting your work to Medical Sciences.

Please find my comments below.

Author: thank you for your review and valuable comments. Please find my responses below.

Line 120: Why only including papers in English or Polish?

Author: This is because most articles are published in English. I used Polish additionally, as it is my native language.

Line 123: Who conducted the study selection process? Normally it should be done by one researcher only and there is only one author in this paper. This applies to the quality assessment process as well. For transparency two authors must be completing that task.

Author: Two researchers were involved in the study selection process. However, I did not include the second person as a co-author, as they did not meet all ICMJE authorship criteria. Instead, I acknowledged their contribution in the Acknowledgements section.

Line 195: The flow chart was not performed according to PRISMA guidelines. For example, you have not added additional records from relevant SRs and / or databases on the top right corner. Also, why was 1 report not retrieved? At the end of the chart, you will need to present the number of included studies in meta-analysis.

Author: I have used the original PRISMA 2020 flow diagram for new systematic reviews, which included searches of databases and registers only (available at https://www.prisma-statement.org/prisma-2020-flow-diagram). One report was not retrieved due to the unavailability of the full-text article in the accessible databases. I have also added the number of studies included in the meta-analysis at the end of the chart.

Line 197: You have decided to synthesise evidence stemming from RCTs and retrospective studies. Therefore, the statistical analysis must be performed separately given the fact you compound the bias (ie not in the same forest plot).

Author: I have considered this issue. In this case performing a sensitivity analysis separating RCTs from observational studies was not feasible due to the small number of available studies. In some subgroup analyses, only two or three studies were included, and further division by study design would have resulted in an insufficient number of studies to perform a meta-analysis.

Line 286: Heterogeneity is unacceptably high here. Consider using GRADE system for incorporating the level of evidence into the results.

Author: I acknowledge that heterogeneity was high in some analyses, which likely reflects the diversity of study designs, populations, and intervention protocols included in this meta-analysis. Given the clinical and methodological variability across trials, I deliberately used random-effects models and reported prediction intervals to account for between-study variance. Also, I have added some new information to limitations section and GRADE score.

Line 423: How was the overall risk of bias assessed?

Author: The overall risk of bias was assessed separately for randomized and non-randomized studies. For randomized controlled trials, I used the Cochrane Risk of Bias 2.0 (RoB 2) tool, evaluating domains such as the randomization process, deviations from intended interventions, missing outcome data, and outcome measurement. For observational studies, the Newcastle-Ottawa Scale was applied, assessing selection, comparability, and outcome domains. The overall risk of bias was then summarized qualitatively based on these assessments.

Reviewer 4 Report

Comments and Suggestions for Authors

The topic of this paper is extremely interesting, and I believe it is relevant to the readers of Medical Sciences. Moreover, the paper is exceptionally well structured and written. The introduction clearly describes the current state of the field and the need for such a study. The Materials and Methods section provides a detailed description of the study design, search strategies, eligibility criteria, and statistical analysis. All obtained results are presented in a highly clear and organized manner, making it easy to understand the main findings of the research. The results are also appropriately discussed in the Discussion section.

There are only a few minor issues that the author should address before publication:

  1. Figure 20 is of somewhat lower quality and should be replaced with a higher-quality image.
  2. Figures 2–19 are excellent, but to enhance the visual appeal of the paper, it might be preferable to present the forest plots in color.
  3. I suggest that the author reconsider the keywords; they are currently quite long (e.g., osteonecrosis of the femoral head), and two of them (osteonecrosis of the femoral head and core decompression) are repeated from the title.

Author Response

The topic of this paper is extremely interesting, and I believe it is relevant to the readers of Medical Sciences. Moreover, the paper is exceptionally well structured and written. The introduction clearly describes the current state of the field and the need for such a study. The Materials and Methods section provides a detailed description of the study design, search strategies, eligibility criteria, and statistical analysis. All obtained results are presented in a highly clear and organized manner, making it easy to understand the main findings of the research. The results are also appropriately discussed in the Discussion section.

Author: I would like to sincerely thank you for your constructive review and your kind, encouraging words. Please find our point-by-point response below.

There are only a few minor issues that the author should address before publication:

  1. Figure 20 is of somewhat lower quality and should be replaced with a higher-quality image.

Author: Thank you for pointing this out. I have added a figure with improved resolution. It is a small change, but I hope it will be sufficient.

  1. Figures 2–19 are excellent, but to enhance the visual appeal of the paper, it might be preferable to present the forest plots in color.

Author: Thank you. Unfortunately, the software used by the statistician allows exporting these forest plots only in black and white.

  1. I suggest that the author reconsider the keywords; they are currently quite long (e.g., osteonecrosis of the femoral head), and two of them (osteonecrosis of the femoral head and core decompression) are repeated from the title.

Author: Good point. I have modified our keywords accordingly.

Round 2

Reviewer 3 Report

Comments and Suggestions for Authors

Dear authors,

Thank you for submitting your revised paper to the Journal.

You have now addressed my original comments and the paper merits publication.